# Targeting the Metabolic Adaptation of Metastatic Cancer

**DOI:** 10.3390/cancers13071641

**Published:** 2021-04-01

**Authors:** Josep Tarragó-Celada, Marta Cascante

**Affiliations:** 1Department of Biochemistry and Molecular Biomedicine, Institute of Biomedicine of Universitat de Barcelona (IBUB), Faculty of Biology, Universitat de Barcelona, 08028 Barcelona, Spain; jtarragocelada@ub.edu; 2Centro de Investigación Biomédica en Red de Enfermedades Hepáticas y Digestivas (CIBEREHD), 28020 Madrid, Spain; 3Metabolomics Node at Spanish National Bioinformatics Institute (INB-ISCIII-ES-ELIXIR), Institute of Health Carlos III (ISCIII), 28029 Madrid, Spain

**Keywords:** colon cancer, metabolism, metastasis

## Abstract

**Simple Summary:**

The search for new therapeutic opportunities to target cancer metastasis is crucial for the improvement of cancer treatment. One of the characteristics of tumoral and metastatic cells is the capacity to reorganize their metabolism, together with the ability to grow faster, migrate and form new tumours in distant sites. Therefore, the pharmaceutical inhibition of metabolic pathways represents a promising strategy to specifically target metastatic cells, especially in colorectal cancer metastasis.

**Abstract:**

Metabolic adaptation is emerging as an important hallmark of cancer and metastasis. In the last decade, increasing evidence has shown the importance of metabolic alterations underlying the metastatic process, especially in breast cancer metastasis but also in colorectal cancer metastasis. Being the main cause of cancer-related deaths, it is of great importance to developing new therapeutic strategies that specifically target metastatic cells. In this regard, targeting metabolic pathways of metastatic cells is one of the more promising windows for new therapies of metastatic colorectal cancer, where still there are no approved inhibitors against metabolic targets. In this study, we review the recent advances in the field of metabolic adaptation of cancer metastasis, focusing our attention on colorectal cancer. In addition, we also review the current status of metabolic inhibitors for cancer treatment.

## 1. Introduction

Cancer metabolic reprogramming is nowadays considered an important hallmark of cancer, receiving every time more attention from cancer researchers and oncologists. However, almost a hundred years ago, Otto Warburg was the first to observe that cancer cells had an increased glucose uptake and lactate production, even in the presence of oxygen [1]. Later, many other metabolic alterations in other pathways have been observed in cancer cells. In the last decade, there has also been an effort to correlate these metabolic changes with the more invasive and metastatic phenotypes of cancer cells, especially in the case of breast cancer metastasis [2,3,4]. Therefore, the need for further research to elucidate the metabolic implications of the metastatic process is crucial for the development of new and more efficient therapies.

The metabolic pathways commonly altered in cancer are related to the main sources of energy and building blocks for sustaining survival and growth of cancer cells [5]. Thus, the Warburg effect and glutamine addiction are two of the main metabolic adaptations associated with cancer [6,7]. Glutamine has a critical role in tumorigenesis as a source of carbon, nitrogen for energy and biosynthetic pathways as well as antioxidant pathways [8]. Other amino acids such as asparagine, arginine, and cysteine have also been observed to be essential for some types of cancer [9,10,11], as well as serine and glycine, which feed one-carbon metabolism pathway generating precursors for the synthesis of nucleotides, DNA methylation processes and redox homeostasis [12]. Altered lipid metabolism is also important as fatty acids play a role in the structural, energetic and signalling demands of cancer cells [13]. Regarding mitochondrial activity, the centre of metabolic pathways, it has been observed that both impaired mitochondria [14] and overactivated mitochondria could be an advantage for cancer cells [15]. Moreover, mutations in TCA cycle enzymes are common in gliomas or leukaemia, generating high levels of what are called oncometabolites such as D-2-hydroxyglutarate, fumarate or succinate that contribute to tumour development through epigenetic regulation [16,17,18,19].

The relationship between all these metabolic changes and other features of cancer has also been largely studied. [20]. The metabolic reprogramming observed during tumour development is driven by altered signalling pathways, resulting in oncogene-directed nutrient uptake and intracellular reprogramming [20]. However, at the same time, alterations in metabolite levels or metabolic enzymes can modulate signalling pathways, causing metabolite-directed changes in cell behaviour and function [5]. Therefore, metabolic and signalling pathways are completely linked and interconnected during cancer development. Some of the signalling pathways that are involved in metabolic reprogramming are PI3K/Akt, MAPK/RAS, MYC, Wnt/β-catenin and HIF1α, among many others [21,22,23,24,25] (Figure 1).

At advanced stages of cancer development, certain cells with cancer stem cell phenotype acquire the capacity to invade the surrounding tissues, intravasate to circulation and finally arrest a new organ or tissue, which is known as the metastatic process [26]. The cells that initiate new tumours in distant sites are considered to be metastatic stem cells. Metastatic tumours are usually resistant to current therapies and constitute the leading cause of cancer death [27]. The mechanisms that drive metastatic development and progression are still poorly understood. However, the genetic alterations on the origin of metastasis are less than those causing tumorigenesis, suggesting that other events such as epigenetic modifications or metabolic reprogramming are more likely to explain the metastatic process [28].

## 2. Metabolic Reprogramming Underlying Metastasis

### 2.1. Glucose Metabolism and Epithelial-Mesenchymal Transition

The metabolic reprogramming associated with the metastatic process is still widely unknown, although some researchers have been working in that specific field during the last few years. The first step of metastasis, which is the transition of the metastatic stem cells from an epithelial to a mesenchymal state, is observed to be accompanied by metabolic changes that frequently are similar to the metabolic reprogramming associated with tumorigenesis and tumour growing [29].

Glycolysis and glycolytic enzymes are enhanced when cells undergo epithelial-mesenchymal transition (EMT), as it happens in tumour phenotypes compared to healthy cells. Energetic and biosynthetic precursors demand in metastatic cells are even more important than in differentiated non-tumorigenic cancer cells. Besides, glycolytic enzymes have also non-metabolic functions related to EMT. For example, GAPDH, G6PD, LDHA and PKM2 have been identified to colocalise with 14-3-3ε protein, gelsolin and G protein β1 subunit at invadopodia [30]. On the other hand, the loss of the gluconeogenetic enzyme fructose 1,6-bisphosphatase (FBP) is associated with E-cadherin loss through Snai1 signalling [31]. Loss of FBP also implies higher flux through the glycolytic enzyme PFK, which is one of the enzymes that regulate glucose metabolism. Another example is PKM2, which is translocated to the nucleus and induce EMT through the upregulation of both ZEB2 and Snai1.

Furthermore, the relationship between metabolic reprogramming and EMT is also explained by other factors that induce both processes, which is the case of TGFβ, TNFα signalling pathways as well as HIF1α. Induced by low oxygen levels in the less vascularised tumour zones or else by certain oncometabolites, HIF1α induces EMT-related genes such as Snai1, Slug, and Twist and regulates the autocrine mobility factor (AMF), which is the secretion form of the glucose phosphate isomerase that also mediates EMT [32]. In turn, HK2, ALDOA, and PKM2, which are activated by HIF1α, are stimulated also by EMT transcription factors, creating a positive feedback loop [33,34] (Figure 2).

In the final steps of glycolysis, the overexpression of enzymes that catalyse lactate production from pyruvate, as well as its transport such as LDH5, MCT1, and MCT4 also correlate with migration capacity [35]. The acidification of the surrounding extracellular environment, also caused by the overexpression of carbonic anhydrase IX (CAIX), help to the first steps of the metastatic process. The effects of an acidic extracellular pH include angiogenesis stimulation, adherens junctions dissociation, extracellular matrix remodelling through activation of hydrolases and MMPs [36,37]. Moreover, acidification does not only occur in the core and hypoxic regions of the tumour but also in the invasive front, inducing an RNA splicing program that helps invasion [38]. The invasive front is not only formed by metastatic cells but also CAFs help in this process and induce invasion by secreting soluble factors that promote nuclear translocation of PKM2, HIF1α activation and ends up with upregulation of ZEB2 and Snai1 through miR-205 [39]. Furthermore, in a model of ovarian cancer cells, CAFs help to mobilise the tumour cell glycogen through IL-6, CXCL10, and CCL5, promoting proliferation, invasion and metastasis [40].

### 2.2. Mitochondrial Metabolism and EMT

The evidence about mitochondrial alterations related to the invasion capacity and EMT is even more contradictory than the one related to tumorigenesis and cancer cell growth. An example of that is the role of the transcription factor PGC-1α that stimulates mitochondrial biogenesis. On one hand, in breast cancer, PGC-1α correlates with the formation of distant metastasis and it is observed to be overexpressed in invasive cells, promoting mitochondrial biogenesis, and enhanced mitochondrial activity [41], as well as overall bioenergetic capacity, flexibility and drug resistance [42]. On the other hand, PGC-1α is observed to suppresses melanoma and prostate metastasis by a different pathway from biogenesis. PGC-1α interacts with ID2 and inactivates the transcription factor TCF4, which, in turn, inactivates genes related to metastasis such as integrins [43,44].

Results about mitochondrial respiration associated with EMT are also contradictory. Both upregulation [45] and downregulation [46] of the electron transport chain have been observed to be related to the acquisition of mesenchymal phenotype and invasive properties. Correspondingly, mitochondrial ROS production may induce EMT and migratory capacities through the activation of Src and Pyk2 [47], although excessive ROS can lead to cell death, being therefore equally important the expression of SOD2 controlled by ZEB2 and NF-κB that would maintain ROS at levels that induce a mesenchymal phenotype [48]. Moreover, the overexpression of ME1, an NADPH producing enzyme, has been correlated with hepatocellular carcinoma malignancy [49]. In cancers that have deregulated TCA cycle enzymes, accumulation of succinate, fumarate, or D2HG cause HIF1α stabilisation and epigenetic modifications that lead to overexpression of EMT transcription factors. For example, ZEB1 overexpression in the case of IDH-mutated gliomas or leukaemias are induced by D2HG levels [50]. Finally, other TCA cycle enzymes that are related to EMT are SDH5, which acts as a tumour suppressor gene and its loss promotes metastasis in lung cancers interacting with GSK-3β and inducing the inhibition of β-catenin [51].

### 2.3. Other Pathways Supporting EMT

Alterations in amino acid metabolism, such as enhanced glutamine uptake and glutaminolysis, have also been related to EMT and metastasis [52,53]. Another amino acid that could have similar roles to glutamine in the metastatic progression is asparagine, whose metabolization by L-asparaginase and also synthesis by asparagine synthase are correlated with invasive and metastatic potential in breast cancer cells [54]. Proline is another example of an amino acid that has been postulated to be important for metastasis formation, as its catabolism through PRODH supports proliferation in the 3D culture of breast cancer cells [55].

Lipid metabolism has also been observed to be implicated in invasion and metastasis. The storage of produced lipids and cholesterol in lipid droplets and destabilisation of lipid rafts driven by FASN is associated with aggressiveness and maintenance of a mesenchymal state trough the induction of VEGF and TFGβ signalling [56,57]. However, diminished fatty acid synthesis could also be an advantage for metastatic cancer cells as it facilitates the availability of acetyl-CoA, implicated in both direct acetylation and acetylation of histones that control genes implicated in EMT such as ZEB1/2 or vimentin [58,59,60,61]. This is consistent with the observation that histone deacetylase inhibitors induce EMT genes in prostate cancer [62]. Fatty acid uptake and lipogenesis (genes such as *CAV1*, *CD36*, *MLXIPL*, *CPT1C*, *CYP2E2*) are also associated with EMT of multiple cancer types and metastasis and poor prognosis [63,64].

### 2.4. Antioxidant Metabolism in Circulating Tumour Cells

When cells finally reach circulation, they become detached. Normal detached cells compromise glucose uptake and result in depressed mitochondrial potential and decreased ATP levels. However, in circulating tumour cells, constitutively active Akt or other signalling pathways are observed to prevent a decrease in ATP levels triggered by the loss of cellular attachment [65,66]. Detachment also causes an apoptotic cell death called anoikis [67] induced by high levels of ROS. Specifically, it is proved in a model of melanoma metastasis in vivo that high ROS levels and oxidative stress limited the observed distant metastasis [68]. Therefore, enhanced antioxidant mechanisms allow the CTC to survive in circulation [69] (Figure 3).

For example, anchorage-independent cells are observed to be dependent on cytosolic IDH1 and produced citrate in a reductive carboxylation manner, that entered to the mitochondria and through IDH2 generated NADPH, which is necessary to compensate high ROS levels [70]. The Erbb2 pathway, which is overexpressed in non-transformed mammary epithelial cells, increased survival of these cells by upregulating the pentose phosphate pathway (PPP) and also generating NADPH [66], something that is also observed in metastatic pancreatic ductal adenocarcinoma cells, which displayed a dependence on the oxidative branch of PPP [71]. Concurrently, glucose oxidation is diminished with matrix detachment, which attenuates the ROS levels that can be produced from the mitochondrial metabolization of glucose [72]. Another antioxidant mechanism activated in CTCs is the increased expression of the activating transcription factor 4 (ATF4) and the nuclear factor-erythroid 2 related factor 2 (Nrf2) that induces heme oxygenase 1 (HO-1) that in turn help to decrease the concentration of prooxidant heme [73]. An in vivo research proved that antioxidant mechanisms favour metastatic cells and specifically circulating tumour cells as mice bearing melanoma and treated by N-acetylcysteine (NAC) developed more metastasis [74].

### 2.5. Metabolic Flexibility in the Metastatic Site

Cancer cells release certain factors in the blood that favour a premetastatic niche in the place of the metastatic site. A metabolic example of that is the secretion of miR-122 that suppress glucose uptake and downregulates PK in the premetastatic niche, which will increase glucose availability for cancer cells once they reach the metastatic site, among other examples [75,76]. In fact, nutrient availability in the tissue or organ of destiny and the adaptation to it by metastatic cancer cells defines the efficiency of metastasis. This is the case of breast cancer metastasis, as observed in a transcriptomics study that revealed differences in the metabolic gene expression between the different metastatic sites [77].

Regarding metabolic adaptations that are observed to be specific of the organ of origin, there is evidence that breast cancer metastatic cells expressed PPP enzymes differently, being the brain the place with the highest expression [78]. Another example of metabolic reprogramming of cells that metastasise to the brain is an increased capacity of acetate metabolization and oxidation in the TCA cycle. This metabolic feature is also observed in primary tumours from the brain, such as glioblastoma, but not in the primary tumours of other sites [79]. Even more, it is also observed that brain metastatic cells from breast cancer can grow independently of glucose and even overexpress gluconeogenic enzymes [80], acquiring the capacity to catabolise GABA to succinate and overexpressing GABA receptor, transporter and transaminase [81].

When breast cancer cells metastasise specifically to the lung, another study reveals that metastatic cells become dependent on pyruvate carboxylase in response to the pyruvate availability of the lung microenvironment [2]. Antioxidant systems such as peroxiredoxin-2 (PRDX2) are also upregulated in metastatic cells in the lungs as the lung is one of the organs with higher oxygen concentration that metastatic cells have to face [82]. Proline catabolism by PRODH also supports metastasis of breast cancers to the lung [55].

When the site of metastasis is the liver, breast cancer cells reduce mitochondrial metabolism as a response to the low oxygen levels that exist in some parts of the liver by overexpressing PDK1 that inhibits PDH and reduces oxidative phosphorylation [83]. The same study indicates that this pattern was unique for liver metastasis compared to breast cancer that metastasised to the lung or bones.

Regarding bone metastasis, breast cancer cells show increased serine biosynthesis through the expression of phosphoglycerate dehydrogenase (PHGDH), phosphoserine aminotransferase, (PSAT1), and phosphoserine phosphatase (PSPH). This is probably occurring because serine is released and required for the formation of osteoclasts that would help bone metastasis through osteolysis [84]. Another metabolite that is observed to fuel osteoclasts through the MCT1 is lactate that breast cancer cells produce in excess through the Warburg effect and released through monocarboxylate transporter 4 (MCT4) [85].

Finally, in lymph node metastasis, fatty acid oxidation is observed to acquire an important role. Enzymes from fatty acid oxidation are overexpressed by the transcriptional coactivator yes-associated protein (YAP) that were probably activated by bile acids accumulation in the metastasis of lymph nodes through the vitamin D receptor [86].

## 3. Metabolic Reprogramming in Colorectal Cancer and Metastasis

### 3.1. Metabolic Alterations in the Primary Stages of Colorectal Cancer

In the specific context of colon cancer, it is observed that at the very initial steps of tumorigenesis and progression, cancer-initiating cells present a significant metabolic reprogramming, mainly characterised by increased glycolysis, TCA cycle, cysteine, and methionine metabolism [87]. These changes could be initially mediated by Wnt signalling, one of the first pathways that are found deregulated in the colon cancer cascade. Wnt signalling is also observed to induce activation of pyruvate dehydrogenase kinase 1 (PDK1), which inhibits PDH, suggesting a decreased flux of glucose to mitochondrial respiration [88]. Wnt pathway is also directly activating the lactate/pyruvate transporter MCT1 which contributes to an increased flux of glycolysis and lactate secretion [89]. Another enzyme of the glycolytic pathway, PKM2 is observed to be activated by O-GlcNAcylation and serine phosphorylation [90].

Apart from direct Wnt pathway-related metabolic alterations in the adenoma stage also MYC is observed to be upregulated, probably induced by Wnt, bringing an early metabolic reprogramming including the major biosynthetic pathways such as PPP, nucleotide synthesis, fatty acid synthesis or one-carbon metabolism. At the same time, MYC reduces the expression of genes related to mitochondrial metabolism such as PGC-1α [91]. Furthermore, in the context of colon cancer initiation, the metabolic pathways that are altered in cancer-initiating cells in the colon epithelia are greatly influenced by the microbiome and vice versa [92]. There are other signalling pathways important in colorectal cancer that could affect metabolic pathways. This is the case of KRAS mutations, which are observed to give some metabolic advantages such as survival under glucose-depletion conditions [93]. Also, PI3K/Akt and p53 are described to be involved in colon cancer metabolic reprogramming [94].

Other metabolic alterations that are observed to be related to colorectal cancer initiation are the overexpression of GLS1, GLUD1, and the mitochondrial aspartate glutamate carrier 2 (SLC25A13), which are associated with tumour growth and poorer outcome in colon cancer [93,95]. Many metabolic alterations in the initial steps of colorectal cancer also occur in lipid metabolism. β-oxidation is another of the specific signatures observed in colon cancer cell lines but not in other types of cancer [96]. More changes in lipid metabolism include increased fatty acid synthesis through FASN, increased elongation of saturated fatty acids, desaturation, and polyunsaturation [97].

### 3.2. Metabolic Alterations in Metastatic Colorectal Cancer

As explained before, the metabolic alterations underlying metastatic colorectal cancer are much less studied than other cancer types such as breast cancer. However, some relevant studies compared primary tumour and metastatic colon metabolism and found that expression of ACSL and SCD correlates with poor clinical outcome inducing energetic capacity and invasive and migratory characteristics [98]. Another study of clinical relevance found an increased expression of GLUT3 and PKM2 in metastatic colon cancer cells through the YAP pathway [99]. In fact, the YAP pathway is also associated with lymph node metastasis through the upregulation of fatty acid oxidation [86].

Other studies have discovered some of the metabolic properties that would allow colorectal cancer cells to survive in circulation and initiate metastasis in the liver: Detached colorectal CTCs overexpressed enzymes of the fatty oxidation pathway such as CPT1A, the rate-limiting enzyme of this pathway. Such enzymes are found to be essential for anoikis resistance and survival of CTCs from the colon [100]. Furthermore, cystine import and folate metabolism have been discovered to be also essential for lymph node and liver metastatic colorectal cancer cell lines [101].

Finally, some studies found a characteristic pattern in colorectal cancer cells that metastasised to the liver (Figure 4). This is the case of the induction of creatine phosphorylation in the extracellular environment by secreting creatine kinase B (CKB) and then import the resulting phosphocreatine through the SLC6A8 transporter and use it as a fuel to generate ATP [102]. Another example is the production of thrombopoietin (TPO) by hepatic cells that activates lysine catabolism through the TPO receptor CD110 expressed in CTCs from the colon. Lysine catabolism generates Acetyl-CoA necessary for LDL receptor-related protein 6 (LRP6) acetylation that in turn leads to Wnt signalling, which promotes self-renewal of colorectal cancer tumour cells that colonise the liver. It also generates glutamate that contributes to glutathione formation [103]. Finally, metastatic cells secrete miR-122 to downregulate PKM2 in hepatocytes and increase glucose availability [75]. In colorectal cancer, miR-122 is found to be upregulated and promote ALDOA overexpression [104]. Precisely, another isoform of this enzyme, ALDOB, is found to enhance fructose metabolism in colorectal metastatic cancer cells as a response to elevated fructose concentration that is present in the liver [4].

## 4. Targeting Metabolic Adaptation

### 4.1. Metabolic-Based Therapeutic Strategies in Cancer and Metastasis

Metabolic reprogramming has been exploited for cancer therapy almost since the beginning of chemotherapy, with the discovery of aminopterin for the treatment of childhood acute lymphoblastic leukaemia [105] and nowadays many metabolic enzyme inhibitors are approved or being explored for cancer therapy (Table 1 and Figure 5). Aminopterin was the first of one of the main classes of chemotherapeutics, antimetabolites, and was the precursor of methotrexate and pemetrexed, which are drugs currently used for many types of cancer. Specifically, these drugs are also called antifolates or folate analogues as they act inhibiting enzymes of folate pathway that are involved in nucleotide synthesis, mainly dihydrofolate reductase (DHFR), although not all of them have the same targets. Since then, it is widely recognised that folate metabolism is one of the main overactivated pathways in cancer, although it is also a crucial pathway for stem cell metabolism and immune system precursors. For that reason, there are many side effects and cytotoxicity related to bone marrow, intestinal crypts and hair follicles affectation [106].

Another commonly used antimetabolites are those inhibiting the nucleotide synthesis directly in biosynthetic pathways of purines and pyrimidines, which are also overactivated in cancer cells. There are purine analogues (6-mercaptopurine and 6-thioguanine), which target amidophosphoribosyltransferase (ATase), and pyrimidine analogues (5-fluorouracil and its prodrug capecitabine), which target thymidylate synthase (TS) [107]. There are also inhibitors of dihydroorotate dehydrogenase (DHODH) (brequinar and leflunomide), one of the steps in the pyrimidine nucleotide synthesis pathway. Despite being widely used in chemotherapy they also showed similar cytotoxic effects as antifolates.

Fortunately, many other metabolic-based strategies are approved or in clinical trials that are more specific for cancer cell metabolic reprogramming. This is the case of the treatment of cancers that present mutations in IDH1 or 2, mainly glioblastoma multiforme (GBM) or acute myeloid leukaemia (AML). Specific inhibitors for IDH1 (AG-120 or ivosidenib, IDH305, BAY143602, FT-2102), IDH2 (AG-221 or enasidenib) or both IDH1/2 (AG-881) have been proven to be effective and approved or currently in clinical trials, being enasidenib and ivosidenib the first mutant-IDH FDA-approved for AML patients in 2017. However, not all IDH-mutant tumours have a good response to the treatment or exhibit resistance, therefore, further exploration of the possible drug combinations is needed [108].

Other metabolic targets are related to glucose metabolism and the TCA cycle. Alpha-enolase (ENO1) is one of the enzymes of glycolysis that is shown to be overexpressed and contribute to the Warburg effect and its inhibition using phophonoacetohydroxamate (PhAH) or SF-2312 has been proven to be effective for many cancer cell lines in vitro [109]. Inhibition of ENO1 also activates plasmacytoid dendritic cells in multiple myeloma patients, enhancing the immune response against tumour cells [110]. The inhibition of the last step of glycolysis, the enzyme PK and specifically its isoform M are another of the most explored spots of cancer metabolic reprogramming and its inhibition (by TEPP-46 or AG-348) has proven to be effective in preclinical models of pancreatic cancer, and in combination with the lactate dehydrogenase inhibitor FX-11. One of the central enzymes of carbon metabolism and that is frequently enhanced in cancer cells is pyruvate dehydrogenase (PDH) [111]. Its inhibition tackles the metabolic reprogramming of cancer cells, impairing cell growth and invasion [112,113], and its inhibitor devimistat (CPI-613) is nowadays in phase III trial in patients with AML. Another target that is being explored is the import and export of lactate mediated by monocarboxylate transporter 1 (MCT1), which allows both the secretion of lactate from highly glycolytic cancer cells as well as the utilization of lactate from the tumour microenvironment. AZD3965, an inhibitor of MCT1, decreases tumour aggressiveness by increasing immune cell infiltration and inhibiting lipid biosynthesis. The inhibitor is nowadays in phase I of clinical trials [114]. Finally, metformin, a drug that is used in the treatment of diabetes and is known to inhibit the mitochondrial respiratory chain complex I is nowadays being explored for cancer prevention and treatment, although its benefits are still not clear [115,116].

Glutamine addiction is another metabolic feature of cancer cells that is being exploited for therapy. Targeting GLS 1 activity with the inhibitor telaglenastat (CB-839) is currently in clinical trials for the treatment of renal cell carcinoma or other solid tumours in combination with other drugs [117,118]. However, it has been recently discovered in c-MYC-induced liver tumours that the inhibition of GLS1 is compensated by the expression of GLS2, and even the inhibition of both isoforms would not completely inhibit glutamine catabolism [119]. Another amino acid that is being explored for cancer therapy is serine, especially de novo serine from glucose. The first enzyme of this pathway, phosphoglycerate dehydrogenase (PHGDH) is overexpressed in many types of cancers such as breast, lung and melanoma. Inhibitors of PHGDH have been designed (NCT-503 and CBR-5884) [120], although they present brain cytotoxic effects and its effectivity depends on environmental serine availability. In fact, some cancer types such as intestinal cancer or lymphoma that are not KRAS mutant rely on exogenous serine uptake and dietary restriction of serine and glycine have an anti-tumour effect in mouse models [121].

Other chemotherapeutic treatments related to amino acid metabolism are the depletion of circulating arginine using PEG-BCT-100 or AEB-1102, which are human recombinant arginases or arginine deaminases, respectively. Such therapies are in phase I of clinical studies in combination with immune checkpoint inhibitor antibodies [122], in the context of argininosuccinate synthase 1 (ASS1)—deficient cancers, normally melanoma, lymphoma, glioma or prostate cancers that rely on circulating arginine. The same approach is applied for circulating asparagine using Asparaginase, which is currently used in the first-line treatment of ALL [123] and it is also being tested for solid tumours and nanoparticles carrying the enzyme are also being developed to improve efficiency [124].

Finally, fatty acid metabolism rewiring in cancer is also currently being explored for cancer therapy. Specifically, de novo lipogenesis is a requirement of cancer cells and targeting fatty acid synthase (FASN) using the TVB-2640 FASN inhibitor is nowadays in phase I or II clinical trials for breast, lung and colon cancers [13]. Inhibition of acetyl-CoA carboxylase (ACC), the previous enzyme of the lipogenesis pathway, is also being explored in preclinical models using the inhibitor ND-646 [125].

### 4.2. Metabolic-Based Therapeutic Strategies in Colorectal Cancer

Many metabolic inhibitors are being tested for the treatment of colorectal cancer in in vitro models, in preclinical models and also in clinical trials such as GLS1 or FASN inhibitors. An example of specifically targeting a colorectal cancer metabolic adaptation is 3-bromopyruvate sensitivity, taking advantage that the colorectal cancer cells present overexpression of MCT1, induced by the Wnt pathway [89,126]. In that sense, the drug AZD3965, which targets MCT1, is being tested for colorectal cancer in preclinical models [127]. Furthermore, the presence of acetate and other short-chain fatty acids that could come also from the microbiome is observed to have an important role in cancer prevention. Another example of an inhibitor being tested in colorectal cancer cell lines is WZB117, which targets GLUT1, to overcome 5-fluorouracil resistance [128]. Metformin, which targets the mitochondrial electron transport complex I, is also widely explored in colorectal cancer in both preclinical and clinical trials [129].

Although there are many metabolic alterations observed in colorectal cancer and other cancer types have started to benefit from metabolic strategies, none of these metabolic approaches is being used nowadays in the treatment of colorectal cancer [130]. Therefore, more research is needed for a better comprehension of the metabolic reprogramming in colorectal cancer and metastasis that would allow discovering new metabolic targets or take profit of the inhibitors that are currently used in the clinics for other types of cancer.

## 5. Conclusions

The metabolic adaptation of cancer has been studied since the Warburg effect was discovered in the 1920s. However, in the last decades, there has been an increasing number of studies evidencing that metabolic changes are crucial not only for tumorigenesis and cancer development but also for metastasis. Multiple examples of metabolic alterations underlying the different steps of the metastatic processes have been discovered in recent years. Despite some of the actual chemotherapies already target the metabolism of cancer cells, there is great hope for the development of more effective metabolic inhibitors. The present review displays evidence that there is an urgent need to better understand the metabolic mechanisms underlying colorectal cancer metastasis, and that there is a huge window of therapeutic opportunities targeting metastatic colorectal cancer metabolic plasticity that should be exploited in the future by means of drug combinations that compromise this metabolic adaptability.

## Figures and Tables

**Figure 1 cancers-13-01641-f001:**
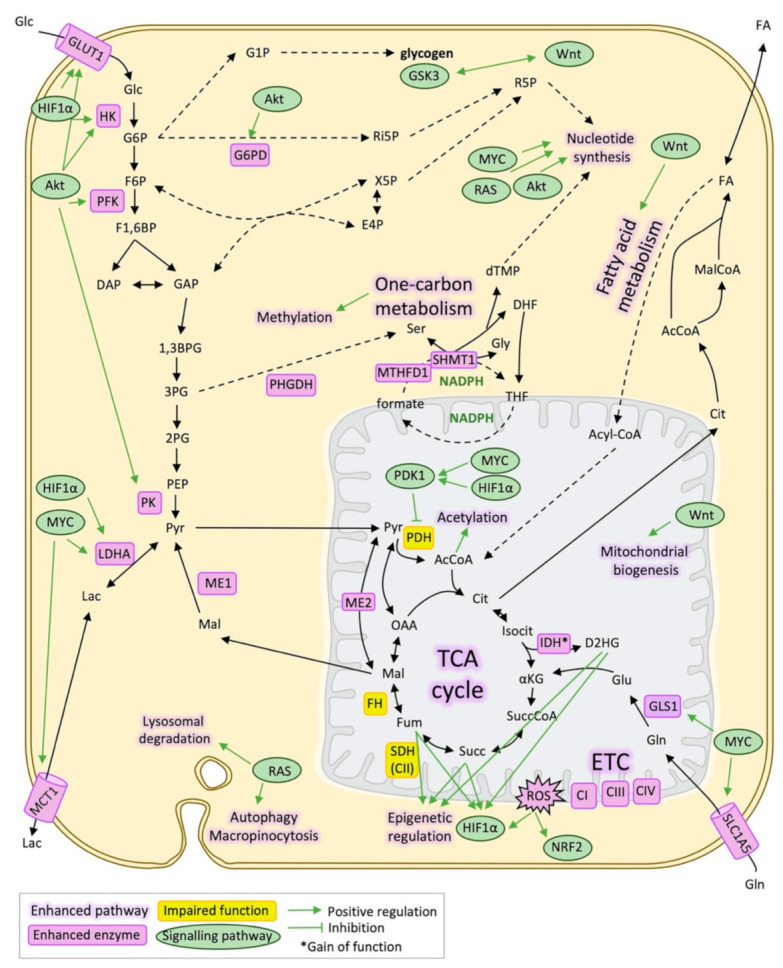
Metabolic reprogramming and signalling pathways in cancer. Schematic representation of the main metabolic pathways and metabolic enzymes altered in cancer and its relationship with signalling pathways that regulate them mainly through transcriptional activation/repression. 1,3BPG: 1,3-bisphosphoglycerate, 2PG: 2-phosphoglycerate, 3PG: 3-phosphoglycerate, AcCoA: Acetyl-CoA, CI: Respiratory complex I, Respiratory complex III, Cit: Citrate, CIV: Respiratory complex IV, DAP: Dihydroxyacetone phosphate, DHF: Dihydrofolate, dTMP: Deoxythymidine monophosphate, E4P: Erythrose 4-phosphate, ETC: Electron transport chain, F1,6BP: Fructose 1,6-bisphosphate, F6P: Fructose 6-phosphate, FA: Fatty acids, FH: Fumarate hydratase, mitochondrial, Fum: Fumarate, G1P: Glucose 1-phosphate, G6P: Glucose 6-phosphate, G6PD: Glucose 6-phosphate 1-dehydrogenase, GAP: Glyceraldehyde 3-phosphate, Glc: Glucose, Gln: Glutamine, GLS1: Glutaminase, kidney isoform, GLUT1: Glucose transporter 1, Gly: Glycine, GSK3: Glycogen synthase kinase-3, HIF1α: Hypoxia-inducible factor 1-alpha, Isocit: Isocitrate, IDH: Isocitrate dehydrogenase, Lac: Lactate, LDHA: L-lactate dehydrogenase A chain, Mal: Malate, MalCoA: Malonyl-CoA, MCT1: Monocarboxylate transporter 1, ME1: NADP-dependent malic enzyme, ME2: NAD-dependent malic enzyme, mitochondrial, NADPH: Nicotinamide adenine dinucleotide phosphate, NRF2: Nuclear factor erythroid 2-related factor 2, OAA: Oxalacetate, PDH: Pyruvate dehydrogenase, PDK1: Pyruvate dehydrogenase lipoamide kinase isoenzyme 1, mitochondrial, PEP: Phosphoenolpyruvate, PHGDH: D-3-phosphoglycerate dehydrogenase, PK: Pyruvate kinase, Pyr: Pyruvate, ROS: Reactive oxygen species, SDH(CII): Succinate dehydrogenase (respiratory complex II), Ser: Serine, SLC1A5: Solute carrier family (neutral amino acid transporter), member 5, Succ: Succinate, SuccCoA: Succinyl-CoA, THF: Tetrahydrofolate, X5P: Xylolose 5-phosphate, αKG: α-ketoglutarate.

**Figure 2 cancers-13-01641-f002:**
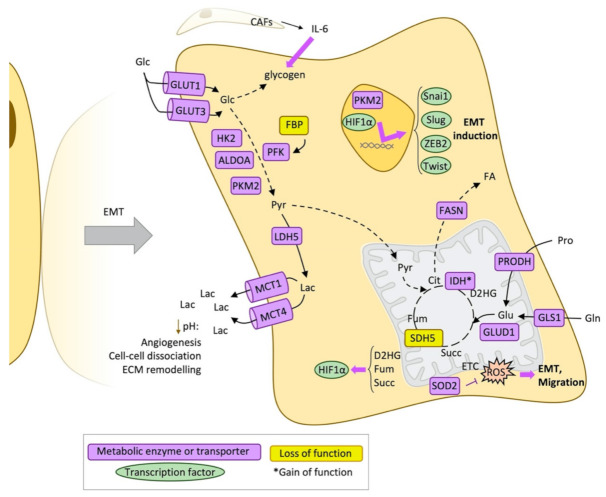
Metabolic reprogramming during epithelial-mesenchymal transition (EMT). Schematic representation of the main metabolic changes described during the epithelial-mesenchymal transition process. ALDOA: Fructose-bisphosphate aldolase A, CAFs: Cancer-associated fibroblasts, Cit: Citrate, D2HG: D-2-hydroxyglutarate, ECM: Extracellular matrix, EMT: Epithelial-mesenchymal transition, ETC: Electron transport chain, FA: Fatty acids, FASN: Fatty acid synthase, Fum: Fumarate, Glc: Glucose, Gln: Glutamine, GLS1: Glutaminase, kidney isoform, GLUD1: Glutamate dehydrogenase 1, mitochondrial, GLUT1: Glucose transporter 1, GLUT3: Glucose transporter 3, HK2: Hexokinase 2, HIF1α: Hypoxia-inducible factor 1-alpha, IDH: Isocitrate dehydrogenase, IL6: Interleukin 6, LDH5: L-lactate dehydrogenase-5, PFK: Phosphofructokinase, PKM2: Pyruvate kinase M2 isoform, Pro: Proline, PRODH: Proline dehydrogenase, Pyr: Pyruvate, ROS: Reactive oxygen species, SDH5: Succinate dehydrogenase subunit 5, SOD2: Superoxide dismutase 2, Succ: Succinate, ZEB2: Zinc finger E-box binding homeobox 2.

**Figure 3 cancers-13-01641-f003:**
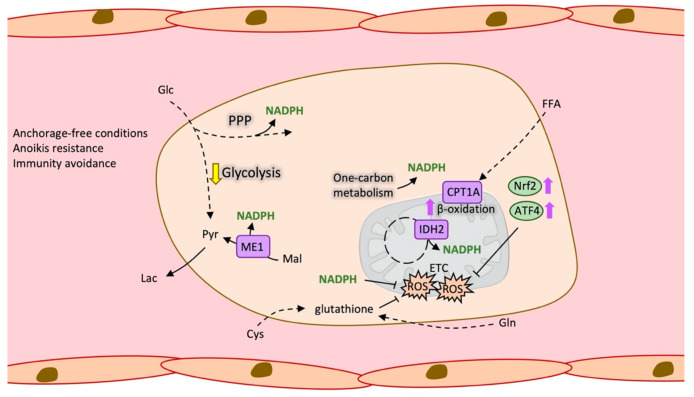
Antioxidant metabolism in circulating tumour cells (CTC). Schematic representation of the metabolic changes that favour CTCs survival under anchorage-free conditions, mainly enhanced ROS. ATF4: Activating Transcription Factor 4, CPT1A: Carnitine palmitoyltransferase 1A, Cys: Cysteine, ETC: Electron transport chain, FFA: Free fatty acids, Glc: Glucose, Gln: Glutamine, Lac: Lactate, Mal: Malate, ME1: NADP-dependent malic enzyme, NADPH: Nicotinamide adenine dinucleotide phosphate, Nrf2: Nuclear factor erythroid 2-related factor 2, PPP: Pentose phosphate pathway, Pyr: Pyruvate, ROS: Reactive oxygen species.

**Figure 4 cancers-13-01641-f004:**
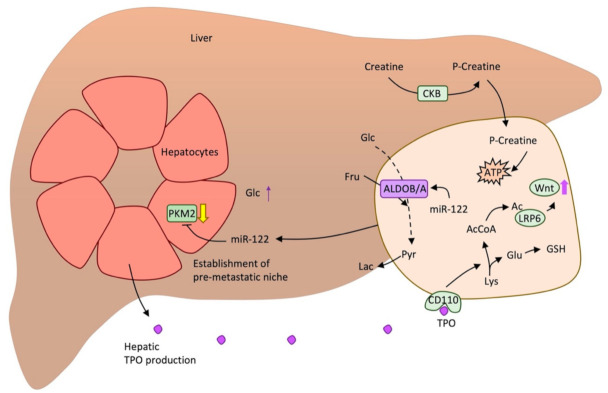
Metabolic flexibility of metastatic colorectal cancer cells colonising the liver. Schematic representation of the known metabolic changes that help cancer cells from the colon to colonise the liver. Ac: Acetate, AcCoA: Acetyl-CoA, ATP: Adenosine triphosphate, CKB: Creatine kinase B, Fru: Fructose, Glc: Glucose, Glu: Glutamate, GSH: Glutathione, Lac: Lactate, LRP6: LDL receptor-related protein 6, Lys: Lysine, PKM2: Pyruvate kinase M2, Pyr: Pyruvate, TPO: Thrombopoietin.

**Figure 5 cancers-13-01641-f005:**
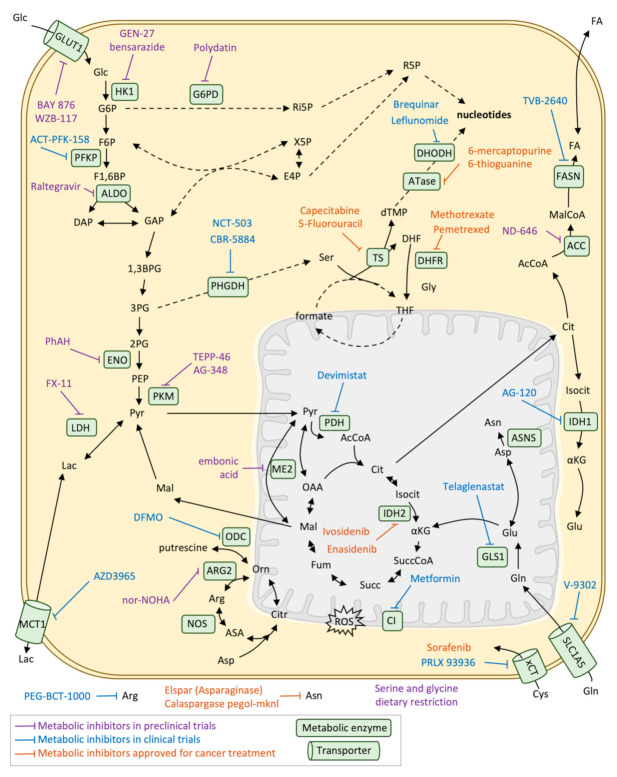
Metabolic inhibitors for cancer treatment. Schematic representation of the main metabolic pathways and the main metabolic inhibitors that are being nowadays explored in preclinical or clinical trials as well as those approved for cancer treatment. 1,3BPG: 1,3-bisphosphoglycerate, 2PG: 2-phosphoglycerate, 3PG: 3-phosphoglycerate, ACC: Acetyl-CoA carboxylase, AcCoA: Acetyl-CoA, Arg: Arginine, ARG2: Arginase 2, Asn: Asparagine, ASNS: Asparagine synthetase, Asp: Aspartate, ATase: Amidophosphoribosyltransferase, CI: Respiratory complex I, Cit: Citrate, Citr: Citrulline, DAP: Dihydroxyacetone phosphate, DHF: Dihydrofolate, DHFR: Dihydrofolate reductase, DHODH: Dihydroorotate dehydrogenase, dTMP: Deoxythymidine monophosphate, E4P: Erythrose 4-phosphate, F1,6BP: Fructose 1,6-bisphosphate, F6P: Fructose 6-phosphate, FA: Fatty acids, FASN: Fatty acid synthase, FH: Fumarate hydratase, mitochondrial, Fum: Fumarate, G6P: Glucose 6-phosphate, G6PD: Glucose 6-phosphate 1-dehydrogenase, GAP: Glyceraldehyde 3-phosphate, Glc: Glucose, Gln: Glutamine, GLS1: Glutaminase, kidney isoform, GLUT1: Glucose transporter 1, Gly: Glycine, IDH2: Isocitrate dehydrogenase mitochondrial, Isocit: Isocitrate, Lac: Lactate, LDH: L-lactate dehydrogenase, Mal: Malate, MalCoA: Malonyl-CoA, MCT1: Monocarboxylate transporter 1, ME2: NAD-dependent malic enzyme, mitochondrial, NADPH: Nicotinamide adenine dinucleotide phosphate, NOS: Nitric oxide synthase, OAA: Oxalacetate, ODC: Ornithine decarboxylase, Orn: Ornitine, PDH: Pyruvate dehydrogenase, PDK1: Pyruvate dehydrogenase lipoamide kinase isoenzyme 1, mitochondrial, PEP: Phosphoenolpyruvate, PHGDH: D-3-phosphoglycerate dehydrogenase, PKM: Pyruvate kinase isoform M, Pyr: Pyruvate, ROS: Reactive oxygen species, SDH(CII): Succinate dehydrogenase (respiratory complex II), Ser: Serine, SLC1A5: Solute carrier family (neutral amino acid transporter), member 5, Succ: Succinate, SuccCoA: Succinyl-CoA, THF: Tetrahydrofolate, TS: Thymidylate synthase, X5P: Xylolose 5-phosphate, xCT: Cysteine/glutamate transporter system xCT, αKG: α-ketoglutarate.

**Table 1 cancers-13-01641-t001:** FDA-approved drugs to target the metabolism of cancer.

Drug	Target	Approved for	Year
Methotrexate	DHFR	Breast cancer, epidermoid cancers of the head and neck, cutaneous T cell lymphoma, lung cancer, etc.	1953
6-mercaptopurine	ATase	ALL	1953
6-thioguanine	ATase	AML, ALL and CML	1966
Elspar	Asparagine	ALL	1978
Capecitabine	TS	Colorectal cancer and breast cancer	1998
5-Fluorouracil	TS	Colorectal cancer and other gastrointestinal cancers, breast cancer, neuroendocrine tumours, thymic cancer, cervical cancer, bladder cancer, hepatobiliary cancer	2000
Pemetrexed	DHFR, TS, GARFT	Malignant pleural mesothelioma, lung cancer	2004
Sorafenib	Many protein kinases, xCT	Renal cell carcinoma, hepatocellular carcinoma and thyroid cancer	2005
Enasidenib	IDH2 mutated	AML	2017
Calaspargase pegol-mknl	Asparagine	ALL	2018
Ivosidenib	IDH1 mutated	AML	2019

ALL: Acute lymphoblastic leukaemia, AML: Acute myeloid leukaemia, ATase: Amidophosphoribosyltransferase, CML: chronic myeloid leukaemia, DHFR: Dihydrofolate reductase, GARFT: Glycinamide ribonucleotide transformylase, IDH1: Isocitrate dehydrogenase cytoplasmatic, IDH2: Isocitrate dehydrogenase mitochondrial, TS: thymidylate synthase, xCT: Cysteine/glutamate transporter system xCT.

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
