# Peer review of "Targeting the Metabolic Adaptation of Metastatic Cancer"

_cancers, 2021, doi:10.3390/cancers13071641_

Round 1

Reviewer 1 Report

The article by Tarrago-Celada and Cascante provides a comprehensive overview about metabolic adaptations in cancer cells in context of metastasis. It also provides a broad overview about current research on metabolic intervention strategies.

Overall, the article is well organised and clearly written. Due to the complexity of the topic it is not possible to go in depth, but it nevertheless, covers the topic comprehensively.

I have only one comment for the authors to think about: In my view, the article covers a much wider area than colorectal cancer (i.e. different cancer types) and I was thus wondering if the title sufficiently reflects its content. 

In summary, I have no concerns and I encourage publication of this article.

Reviewer 2 Report

Major Concerns:

1) In the Simple Summary:

“ One of the characteristics of tumoral and metastatic 14 cells is the capacity to reorganize its metabolism, in order to be able to grow faster, migrate and form 15 new tumours in distant sites.” , in which the phrase of “in order to” was inappropriate because it is not known the role of metabolic alterations in cancer are to drive cell growth since fast growing benign tumors do not present altered metabolism.

2) In the Introduction, lines 42-43:

“The metabolic pathways commonly altered in cancer are related to the main sources of energy and building blocks for the generation of new biomass.”, in which the statement of “for the generation of new biomass” is not sufficient because altered metabolism is more important for sustaining survival of cancer cells.

3) In the Introduction, lines 48-50:

The statement of “Lipid metabolism is also enhanced as a need to increase phospholipid synthesis for cell division, as well as to increase energy production and acetyl formation for protein acetyla- tion” is incorrect, because we still do not understand why cancer cells present enhanced de novo lipogenesis regardless of the availability of exogenous lipid.

4) In the Introduction, line 58-59,

“During tumorigenesis, the overexpression of oncogenes and the repression of tumour suppressor genes are responsible for changes in signaling pathways that ultimately regulate metabolic reactions, among other processes.”, in which the underlined phrase is incorrect because we do not know where altered metabolism is a cause or a consequence of genetic alterations.

Minor issues:

1) the word “evidence” shall remain to be singular.

2) statements in Line 116 and 117 are missing citations

3) Line 201, to be precise, please add “metastatic cancer ” into the first sentence to define the “cells”

4) Line 201-203 missing citation

5) Line 246-248 missing citation

Reviewer 3 Report

The author did a great job reviewing on metabolic regulation of colon cancer and how can we target these metabolic pathways to inhibit cancer metastasis. Only two points need to be added to this manuscript. 

  1. glutamine addiction is an important aspect during tumorigenesis. Thus authors need to expand this section.
  2. It will be great to include a table listing all FDA-approved drug to target the metabolism of cancer.
